# Snail Transcriptionally Represses *Brachyury* to Promote the Mesenchymal-Epithelial Transition in Ascidian Notochord Cells

**DOI:** 10.3390/ijms25063413

**Published:** 2024-03-18

**Authors:** Bingtong Wu, Xiuke Ouyang, Xiuxia Yang, Bo Dong

**Affiliations:** 1Fang Zongxi Center for Marine EvoDevo, MoE Key Laboratory of Marine Genetics and Breeding, College of Marine Life Sciences, Ocean University of China, Qingdao 266003, China; wubingtong@stu.ouc.edu.cn (B.W.); ouyangxiuke88@163.com (X.O.); bodong@ouc.edu.cn (B.D.); 2Laboratory for Marine Biology and Biotechnology, Qingdao Marine Science and Technology Center, Qingdao 266237, China; 3MoE Key Laboratory of Evolution and Marine Biodiversity, Institute of Evolution and Marine Biodiversity, Ocean University of China, Qingdao 266003, China

**Keywords:** Snail, *Brachyury*, MET, notochord, *Ciona*

## Abstract

Mesenchymal-epithelial transition (MET) is a widely spread and evolutionarily conserved process across species during development. In *Ciona* embryogenesis, the notochord cells undergo the transition from the non-polarized mesenchymal state into the polarized endothelial-like state to initiate the lumen formation between adjacent cells. Based on previously screened MET-related transcription factors by ATAC-seq and Smart-Seq of notochord cells, *Ciona robusta Snail* (*Ci-Snail*) was selected for its high-level expression during this period. Our current knockout results demonstrated that *Ci-Snail* was required for notochord cell MET. Importantly, overexpression of the transcription factor Brachyury in notochord cells resulted in a similar phenotype with failure of lumen formation and MET. More interestingly, expression of *Ci-Snail* in the notochord cells at the late tailbud stage could partially rescue the MET defect caused by Brachyury-overexpression. These results indicated an inverse relationship between *Ci-Snail* and *Brachyury* during notochord cell MET, which was verified by RT-qPCR analysis. Moreover, the overexpression of *Ci-Snail* could significantly inhibit the transcription of *Brachyury*, and the CUT&Tag-qPCR analysis demonstrated that *Ci-Snail* is directly bound to the upstream region of *Brachyury*. In summary, we revealed that *Ci-Snail* promoted the notochord cell MET and was essential for lumen formation via transcriptionally repressing *Brachyury*.

## 1. Introduction

Mesenchymal-epithelial transition (MET), as the reverse process of epithelial-mesenchymal transition (EMT), has been shown to occur widely in embryonic morphogenesis and organogenesis [1], as well as in the malignant process of tumors for spreading and extravasation to colonize the metastatic site [2]. Cells that undergo MET exhibit dramatic shape changes from the motile and non-polarized mesenchymal state to an immotile and polarized epithelial state, which would form a physical and functional barrier between the organism and the external environment, as well as closed cavities [1,3]. In addition to the establishment of epithelial polarity, increasing evidence has established their presence linked to cell fate changes, metabolic switching, and epigenetic modifications [4].

The key EMT-associated transcriptional repressor Snail was first identified in *Drosophila* embryos [5]. It acted as a boundary repressor to regulate mesodermal differentiation by down-regulating ectodermal gene expression within the mesoderm [6]. Its conserved functions were also presented in insects, urochordates, and vertebrates [7,8]. There are three Snail family proteins in vertebrates, namely Snail1 (Snail), Snail2 (Slug), and Snail3 (Smuc). Snail family proteins are characterized by a SNAG box in their N terminus required for transcriptional repression through recruiting co-repressors such as the histone deacetylase family [9]. The DNA binding domains in the C terminus show specificity for sequences centered on the 5′-CACCTG-3′-core. It has been shown that the vertebrate Snail family members Snail and Slug are involved in the regulation of the EMT processes by down-regulating mesenchymal markers (Fibronectin, Vimentin) and up-regulating cell junction-associated proteins (E-cadherin, Claudin, and Plakoglobin) [10]. However, only one Snail protein was identified in *Ciona* without a SNAG box [7], and its function remains to be elucidated.

The T-box gene *Brachyury* is a core regulatory transcription factor required for the *Ciona* notochord formation [11] and is restricted to be expressed in notochord cells by the action of the transcriptional repressor Snail, which leads to the subdivision of embryonic mesoderm into muscle and notochord lineages, separately [12]. During *Ciona* embryogenesis, ten presumptive notochord cells undergo two divisions, then develop into a rod-like structure composed of 40 loosely packed cells along the midline of the body [13]. Next, at the late tailbud stage (Stages 23 to 25), notochord cells undergo MET, in which the apical domains initially appear at the center of lateral domains of each notochord cell, and then extracellular lumen is deposited between adjacent notochord cells and expanded continuously [14,15]. During this process, each notochord cell transforms into an atypical epithelial cell with two apical domains [16]. However, the underlying mechanism of notochord cell MET is unclear. 

In this study, we found that *Ci-Snail* played an important role in *Ciona* notochord cell MET. Both the knockout of *Ci-Snail* and the overexpression of *Brachyury* in notochord cells resulted in the failure of notochord cell MET, indicating the inverse correlation between *Brachyury* and *Ci-Snail*. We verified that expression of *Ci-Snail* in late tailbud-staged notochord cells triggered MET occurrence via down-regulating *Brachyury* expression, based on the facts that overexpression of *Ci-Snail* could rescue MET defect caused by *Brachyury* overexpression. 

## 2. Results

### 2.1. Snail Plays a Critical Role in Ciona Notochord MET

Our previous works searching for functionally important transcription factors related to notochord lumen formation using ATAC-seq have identified *Ci-Snail* for its open chromatin region [17] (Appendix A). The subsequent Smart-Seq of notochord cells revealed *Ci-Snail* expression during the lumenogenesis period (Appendix A). Phylogenetic analysis showed that the *Ci-Snail* was closely related to the vertebrate Snail, but amino acid sequence alignment indicated that *Ci-Snail* lacked the SNAG domain, which was characteristic of vertebrate Snail family members [18] (Appendix A). *Ci-Snail* antibody staining indicated that *Ci-Snail* was localized in the nuclei of notochord cells from Stage 23 to 25 when notochord cell MET occurs (Appendix A). To explore whether *Ci-Snail* plays an essential role in notochord lumen formation, knockout of *Ci-Snail* was carried out by CRISPR/Cas9 in notochord cells. The sgRNA targeted the second exon was selected for *Ci-Snail* knockout (*Ci-Snail* KO) after efficiency evaluation of sgRNA using co-electroporation of plasmid *EF1α*>NLS::Cas9::NLS::P2A-mCherry, together with *Ci-Snail* sgRNA or control sgRNA (Figure 1A and Appendix A). Then, notochord-expressing Cas9 through the construction of plasmid *Brachyury* (1 kb)>NLS::Cas9::NLS::P2A-mCherry, together with *Ci-Snail* KO sgRNA or control sgRNA was introduced into *Ciona* embryos to observe *Ci-Snail* KO-induced phenotypes. It was found that embryos with *Ci-Snail* KO displayed lumen deficiency at Stage 25 compared with the control group, which showed a developed lumen (Figure 1B). 

The biological process of notochord lumen formation indicated that notochord cells went through MET in *Ciona* [15,16], which was characterized by apical membrane occurrence and expansion of notochord lumen. We thus further detected whether *Ci-Snail* KO influenced the MET occurrence. Caveolin, as the marker of caveolae, was recruited to the apical membrane of notochord cells during the MET process [19]. We examined the subcellular localization of Caveolin-mCherry in *Ci-Snail* KO notochord cells, and the results showed that Caveolin-mCherry proteins were diffused in the cytoplasm, while it normally localized at the apical membrane of notochord cells (Figure 1C). This result suggested that *Ci-Snail* affected the MET process in the notochord. MET defect in the *Ciona* notochord cell affected the following cell extension, which would be reflected in the tail dimension. We measured the tail length as a quantitative indicator for the *Ci-Snail* KO consequence and found that the embryo tail length was significantly shorter in the *Ci-Snail* KO group compared to the control group (Figure 1D). In conclusion, *Ci-Snail* expression in the notochord cells was essential for notochord cell MET.

### 2.2. Brachyury Overexpression Affects Ciona Notochord Cell MET

In *Brachyury*-deficient early *Ciona* embryos, notochord cell differentiation is severely impaired [20]; in this study, we carried out *Brachyury* overexpression (*Brachyury* OE) to examine the effects of *Brachyury* on notochord development. We have utilized the promoters of genes *KH.L96.34* (*Rab11a*) and *KH.L22.27* (*chitin synthase-like*) to drive the expression of *Brachyury*. These two genes are specifically expressed in notochord cells at the initial tailbud and late tailbud stage [21], respectively, based on the *Ciona* single-cell database (https://singlecell.broadinstitute.org/single_cell) [22] (accessed on 11 October 2021) (Figure 2A). Brachyury-eGFP overexpression driven by both *KH.L96.34* and *KH.L22.27* promoters affected the notochord lumen formation compared with the control group (Figure 2B).

To further explore the influence of *Brachyury* OE on the MET process, we co-electroporated plasmids *KH.L22.27*>Brachyury-eGFP together with *Brachyury* (1 kb)>Caveolin-mCherry to examine the subcellular localization of caveolin proteins. It was found that *Brachyury* OE resulted in caveolin distribution in the cytoplasm other than at the apical membrane at the late tailbud stage, while caveolin was recruited at the apical membrane of notochord cells in the control group (Figure 2C). This result suggested that *Brachyury* OE caused MET defect in the notochord. Taken together, the downregulation of *Brachyury* expression at the late stage of development facilitates notochord cell MET. 

### 2.3. Ci-Snail OE Partially Rescue the Delayed MET Caused by Brachyury OE

Since *Ci-Snail* proteins were essential for notochord MET, we intend to rescue MET defect caused by *Brachyury* OE through overexpression of *Ci-Snail*. The notochord MET defect phenotype was intended through overexpressing *Brachyury* driven by the *KH.L96.34* promoter from the initial tailbud stage. *KH.L22.27* promoter was chosen to drive the expression of *Ci-Snail* from the late tailbud stage. To experimentally verify the possibility, we co-electroporated *KH.L96.34*>Brachyury-eGFP and *KH.L22.27*>Snail-tdTomato together into the fertilized eggs and sequentially sampled from stages 23 to 25. It was found that no lumen formed between adjacent notochord cells at stage 23 when Brachyury-eGFP was detectable but no *Ci-Snail* overexpression; embryos with strong Brachyury-eGFP expression but weak *Ci-Snail* overexpression showed small lumen between adjacent notochord cells at Stage 24; in contrast, enlarged lumen between adjacent notochord cells were present with both robust *Ci-Snail* and Brachyury-eGFP expression at Stage 25 (Figure 3A). We further assessed the tail length of *Ciona* embryos. The results showed that embryos with both strong expression of *Ci-Snail* and *Brachyury* had significantly longer tails than groups with only *Brachyury* OE but shorter tails than wild-type embryos (Figure 3B).

### 2.4. Ci-Snail Transcriptionally Represses Brachyury Expression to Regulate Notochord MET

*Ci-Snail* OE could rescue the lumen formation defect caused by *Brachyury* OE, suggesting that *Ci-Snail* functions as a transcriptional repressor. To uncover the regulation of *Ci-Snail* on *Brachyury* expression, we first characterized the temporal expression of *Ci-Snail* and *Brachyury* of wild-type embryos at Stages 24 and 25 by RT-qPCR. Results showed that their expressions were inversely correlated (Figure 4A), which was in accordance with our Smart-Seq of notochord cells at the above stages (Appendix A). In *Ciona* embryos at the early tailbud stage, *Ci-Snail* directly binds the upstream regulatory region of *Brachyury* for transcriptional repression to promote the embryonic mesoderm development into tail muscle [12]. To reveal whether *Ci-Snail* transcriptionally represses *Brachyury* in the notochord cell MET, the expression of *Brachyury* was detected before and after *Ci-Snail* was overexpressed at Stage 24. It was found that the *Ci-Snail* OE down-regulated *Brachyury* significantly (Figure 4B). To validate the direct binding of *Ci-Snail* to the upstream region of *Brachyury* in notochord cells, we performed CUT&Tag-qPCR experiments (Figure 4C). Fold enrichment of the target sequence containing the predicted *Ci-Snail* binding site was significantly greater in the HA antibody incubated group compared to the IgG control group (Figure 4D). Taken together, our results indicated that *Ci-Snail* directly binds to the *Brachyury* upstream region to repress its transcription.

## 3. Discussion

We have presented evidence that *Ci-Snail* plays a crucial role in *Ciona* notochord cell MET process. MET occurrence require the presence of *Ci-Snail* but not overexpressed *Brachyury*. The elevated *Ci-Snail* proteins transcriptionally repress *Brachyury* expression to trigger MET occurrence at late tailbud stage. 

Snail, as an EMT-related transcription repressor, is required for mesoderm and neural crest formation during early embryonic development [7,8]. Our previous work detecting temporal expression of *Ci-Snail* by RT-qPCR showed that it presented high expression during gastrulation (16 °C, 10 hpf) but decreased expression during the early tailbud stage (16 °C, 14 hpf); however, elevated expression again at late tailbud stage, accordingly the MET initiation stage (16 °C, 18 hpf) (Appendix A). The expression pattern indicated the transcriptional reactivation of *Ci-Snail* during the notochord MET process, which was also verified by ATAC-seq analysis (Appendix A). Immunofluorescence analysis of *Ci-Snail* revealed that it could be detected both in notochord cells and tail muscle cells at the late-tailbud stage (Appendix A). The temporal and spatial expression of *Ci-Snail* presented evidence for *Ci-Snail* function in notochord MET event. 

Down-regulation of Snail was usually induced during the MET process [23], and even Snail silencing reversed EMT in prostate cancer cells [24]. Here, we demonstrated that *Ci-Snail* expression was required for MET occurrence, and knockout of *Ci-Snail* resulted in MET defect in *Ciona* notochord cells. The indispensability of *Ci-Snail* in two reverse biological processes may be realized through transcriptionally repressing different genes, which is speculated based on the evidence that Snail targets varied genes to regulate broad spectrum of biological processes, including cell proliferation, immune regulation, and stem cell biology [25,26,27].

The T-box transcription factor Brachyury has been proven to be an essential core gene for posterior mesoderm formation and notochord genesis and differentiation [11], and *Brachyury*-knockdown notochord progenitors survive but adopt neural and mesenchymal fates [28]. But when embryos developed into the late tailbud stage, *Brachyury* expression needed to be repressed by *Ci-Snail* to facilitate the MET process, just like the transcriptional regulation in *Ciona* tail muscle cells, though it resulted in different biological events [12]. Otherwise, *Brachyury* overexpression leads to failure of notochord cell MET. The result that *Ci-Snail* OE could only partially rescue MET failure caused by *Brachyury* OE ascribed to the fact that overexpressed *Ci-Snail* could transcriptionally represses the expression of an endogenous gene, which down-regulated the total level of *Brachyury* in the notochord cells. Thus, we conclude that the expression of *Brachyury* might be delicately regulated during the notochord cell MET process. The inhibition of *Brachyury* expression was reported to facilitate the MET process through the down-regulation of mesenchymal markers (Fibronectin, Vimentin) and up-regulation of cell junction-associated proteins (Plakoglobin) [10], which are conserved regulatory mechanisms in the MET process and might lead to the abnormal localization of caveolin in *Ciona* notochord cells. 

Vertebrate Snail family members all share a similar organization, being composed of highly conserved four to six zinc fingers for E-box binding, and the repressor activity depends on the so-called SNAG domain [7]. It has been shown that some Snail family member does not seem to heterochromatinize the target region to ensure long-range silencing although it can silence neighboring genes efficiently [29,30,31]. *Ci-Snail* lost the SNAG domain just like some *Drosophila* family members and *Caenorhabditis elegans*, whose transcriptional repression activity is mediated through an interaction with co-repressor such as CtBP (Carboxy-terminal Binding Protein) [32]. It is speculated that *Ci-Snail* also conjuncts with CtBP to play the repressor activity because of the CtBP consensus site. The repression mechanism of *Ci-Snail* combined with CtBP or other different co-repressors remains elusive. 

MET plays an essential role in the establishment of epithelial polarity in organogenesis during development [33], as well as in the malignant process of tumors [2]. The present study has verified the crucial role of *Ci-Snail* proteins in triggering MET occurrence, which may provide a new thought for deeply understanding the metamorphosis regulation and cancer treatment. 

In summary, *Ci-Snail* proteins are required for notochord cell MET at late-tailbud embryo of *Ciona*. This process is regulated by elevating *Ci-Snail* expression to transcriptionally repress *Brachyury* (Figure 5).

## 4. Materials and Methods

### 4.1. Animal Breeding and Electroporation Experiments

200–300 *Ciona* adults were harvested from Sangou Bay, Rongcheng City, Shandong Province, China. After 24 h culture, 2–3 adult *Ciona* with stout white sperm glands and gray eggs in the oviduct were selected. Mature eggs and sperm were taken and artificially inseminated at room temperature in the laboratory. To obtain transgenic *Ciona* embryos, fertilized eggs were dechorionated as previously described [34]. The working solution for electroporation was prepared by mixing 60 μg plasmid, 420 μL of 0.77 M D-mannitol, and 300 μL of dechorionated eggs to a final volume of 800 μL. Electroporation was performed using the Gene Pulser Xcell system (BIO-RAD, Hercules, CA, USA) with parameters of voltage 50 V and capacitance 1500 μF. After electroporation, the eggs were transferred to plastic petri dishes coated with 1% agarose and incubated at 16 °C. 

### 4.2. Phylogenetic Analysis and Protein Domain Analysis

The sequence of *Ci-Snail* was downloaded from Aniseed (https://aniseed.fr/) (accessed on 15 September 2021), and other Snail protein sequences including vertebrates, protochordates, and invertebrates were downloaded from NCBI (https://www.ncbi.nlm.nih.gov/) (accessed on 2 September 2021). Phylogenetic analysis was constructed using MEGA-X software (version 10.0, Mega Limited, Auckland, New Zealand) with maximum likelihood method and modeled as JTT + G + I, Sequence comparison of proteins using ClustalW method. CD-Search was used for conserved domain analysis. 

Snail protein ID and protein sequence length information for all species are shown in Appendix A.

### 4.3. Immunofluorescence

Embryos were collected and fixed with 4% paraformaldehyde (PFA) in seawater for 2 h at room temperature, then washed with 0.1% PBST 3 times in 8 h. After being blocked with 10% goat serum (Solarbio, Beijing, China) for 1 h at room temperature, the embryos were incubated with *Ci-Snail* antibodies (made by our lab) diluted in 10% goat serum (Snail, 1:100) for 16 h at 4 °C, then followed a washing procedure as above. Another incubation with the secondary antibodies (Alexa Fluor ™ 555 anti-mouse IgG, 1:300, Thermo Fisher, Waltham, MA, USA) and the following washing was performed the same way as the *Ci-Snail* antibodies. Eventually, embryos are mounted with DAPI and imaged by Zeiss LSM900 or LSM980 confocal microscope.

### 4.4. Plasmid Construction

PCR amplification was performed using Phanta Max Super-Fidelity DNA polymerase (Vazyme, Nanjing, China). DNA was extracted with FastPure Gel DNA Extraction Mini Kit (Vazyme, Nanjing, China). DNA fragments were ligated by ClonExpress II One Step Cloning Kit (Vazyme, Nanjing, China). DNA sequencing was synthesized at Sangon Biotech (Shanghai, China).

The 4 kb promoter sequence of notochord-specific gene *KH.L22.27* was amplified by PCR and ligated into the vector pEGFP-N1 (Clontech, Beijing, China) to generate *KH.L22.27*>eGFP. *KH.L22.27*>Snail-eGFP and *KH.L22.27*>Brachyury-eGFP were constructed by subcloning the coding sequence of *Ci-Snail* and *Brachyury* into vector *KH.L22.27*>eGFP at BamH1 site, respectively. *KH.L22.27* was ligated into the vector ptdTomato-N1 (Clontech, Beijing, China) to generate *KH.L22.27*>tdTomato. *KH.L22.27*>Snail-tdTomato was constructed by subcloning *Ci-Snail* into *KH.L22.27*>tdTomato between EcoR1 and BamH1 sites.

The 4 kb promoter sequence of notochord-specific gene *KH.L96.34* was amplified and ligated into the vector pEGFP-N1 to generate *KH.L96.34*>eGFP. *KH.L96.34*>Brachyury-eGFP was constructed in the same way as *KH.L22.27*>Brachyury-eGFP. 

PCR programs during plasmid construction were as follows: pre-denaturation at 95 °C for 3 min; 35 cycles of denaturation at 95 °C for 15 s, annealing at 58 °C for 15 s and extension at 72 °C (1 kb/30 s); then extension at 72 °C for 5 min.

Primers for plasmid construction are shown in Appendix A.

### 4.5. Gene Knockout by CRISPR/Cas9 

The single guide RNA (sgRNA) sequences targeted *Ci-Snail* were designed using the online website CRISPRdirect (http://crispr.dbcls.jp/) (accessed on 18 May 2022). The sgRNA was synthesized and cloned into the vector *Ci-U6*>sgRNA (F+E) (Addgene, Watertown, MA, USA 59986) for expression, and the sequence of control sgRNA was designed as previously described [35]. Two plasmids, *Brachyury* (1 kb)>NLS::Cas9::NLS::P2A-mCherry and *EF1α*>NLS::Cas9::NLS::P2A-mCherry, were employed for Cas9 expression. 

The red fluorescence-emitting embryos were collected, and genomic DNA was extracted for PCR. The PCR products were purified by the Gene JET Gel Extraction Kit (Thermo Fisher, Waltham, MA, USA). A total of 200 ng PCR products were diluted to 17 μL by ddH_2_O and mixed with 2 μL T7 reaction buffer (Vazyme, Nanjing, China) and incubated with 1 μL T7 endonuclease (Vazyme, Nanjing, China) at 37 °C for 30 min. The fragments were analyzed with agarose gel electrophoresis. The PCR products were sequenced and analyzed by synthego (https://ice.synthego.com/) (accessed on 14 November 2023).

Primers for sgRNA are shown in Appendix A.

### 4.6. RNA Extraction and RT-qPCR 

The total RNA was extracted by RNAiso plus reagent (Takara, Beijing, China). The integrity and quality of total RNA was evaluated by agarose gel electrophoresis and Nanodrop one (Thermo Fisher, Waltham, MA, USA). The first-strand cDNA was synthesized using 1 μg total RNA per 20 μL reaction system with HiScript II QRT SuperMix for qPCR Reverse transcription Kit (Vazyme, Nanjing, China). 

The primers for real-time quantitative PCR (RT-qPCR) of the *Ci-Snail* and *Brachyury* were designed with Beacon Designer (Version 7.91, Premier Biosoft, San Diego, CA, USA). RT-qPCR was performed using the SYBR Green PCR Master Mix (Vazyme, Nanjing, China) on Light Cycler 480 (Roche, Basel, Switzerland). The qPCR procedures were as follows: 95 °C for 30 s (permutability); 45 cycles at 95 °C for 10 s, 58 °C for 30 s (cycle reaction); 95 °C for 15 s, 60 °C for 60 s, and 95 °C for 1 s (dissolution curves). The relative expression of *Ci-Snail* and *Brachyury* was normalized with U6 as a reference and calculated using the 2^−ΔΔCt^ method.

Primers for RT-qPCR are shown in Appendix A.

### 4.7. CUT&Tag-qPCR

Fluorescent embryos expressing recombinant protein *Ci-Snail*-HA were digested into single-cell suspension with 1% trypsin. In the CUT&Tag procedure, cells were bound and held on Concanavalin A-coated magnetic beads. After permeabilization by the Digitonin (nonionic decontaminating agent), cells were incubated sequentially with the primary antibody (Mouse anti-HA-Tag mAb, AE008, Abclonal, Wuhan, China) against the *Ci-Snail*-HA protein and the corresponding secondary antibody (goat anti-mouse IgG H&L, ab6708, abcam, Cambridge, UK), and fusion protein of Protein-A/G and Tn5 transposase for genome localization, cleavage and adapter sequences insertion nearby the target protein. The fragmented DNA was retrieved as a template for subsequent RT-qPCR to amplify products containing *Ci-Snail* binding sequences. The fold enrichment of the target sequence was calculated through the 2^−ΔΔCt^ method.

### 4.8. Statistical Analysis

All graphs in this study were produced using GraphPad Prism (Version 8.0.2, GraphPad, San Diego, CA, USA). Statistical analysis was performed using a *t*-test, and a *p-*value less than 0.05 was considered statistically significant, less than 0.01 was a highly significant difference, and less than 0.001 was an extremely significant difference.

## Figures and Tables

**Figure 1 ijms-25-03413-f001:**
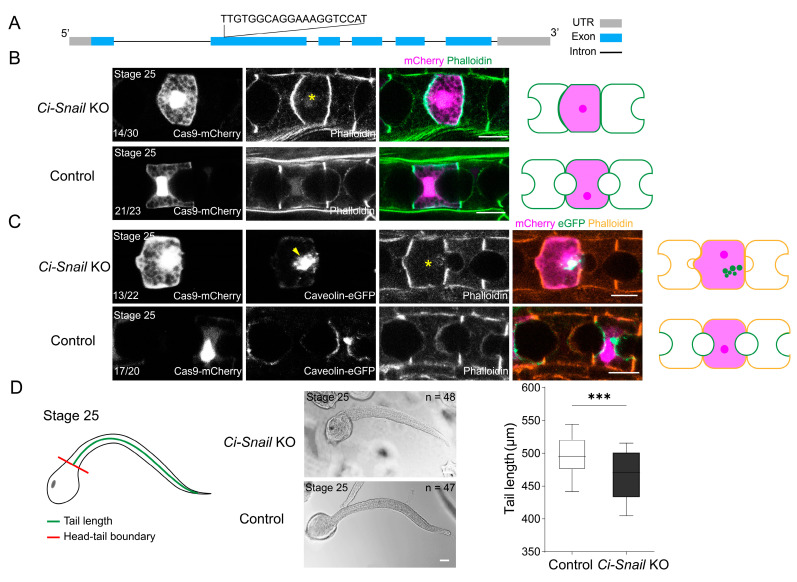
*Snail* KO in the notochord leads to abnormal MET in *Ciona*. (**A**) *Ci-Snail* KO sgRNA was designed to target the second exon. (**B**) *Ci-Snail* KO in notochord cells resulted in failure of lumen formation. The yellow asterisk “*” indicates *Ci-Snail* KO cells. The green represents the cell membrane boundary, and the dark purple and purple represent nuclei/cytoplasm with Cas9 expression, respectively. (**C**) *Ci-Snail* KO in notochord cells resulted in a dispersed distribution of caveolin. The yellow asterisk “*” indicates *Ci-Snail* KO cells and the yellow arrowhead indicates caveolin localization. The orange, green, and purple represent the cell membrane boundary, caveolin localization, and the nuclei/cytoplasm with Cas9 expression, respectively. The schematic images on the right side illustrate the phenotypes of the transgenic notochord cells; the numbers at the bottom left of the images indicate the percentage of notochord cells with the illustrated phenotype. (**D**) Comparison of the tail length of the *Ci-Snail* KO group with the control group. The measurement is illustrated with a cartoon embryo. The tail length is illustrated with a green line, and the junction between the head and tail is a red line. The numbers in the upper right of the DIC images indicate the quantity of the embryos counted. Statistical analysis was performed using *t*-tests. “***” means the difference between the two groups is extremely significant (*p* < 0.001). All of the scale bars represent 10 μm.

**Figure 2 ijms-25-03413-f002:**
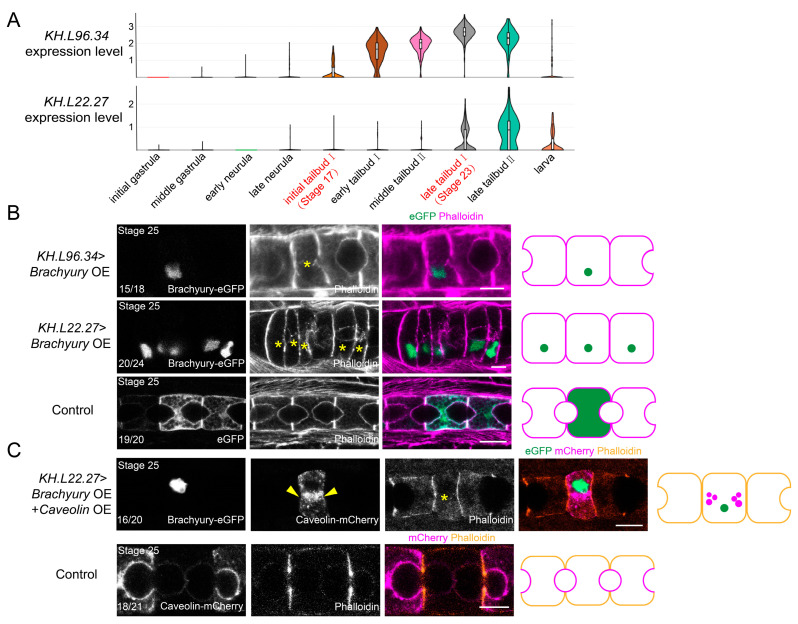
*Brachyury* OE leads to abnormal MET in *Ciona* notochord. (**A**) *Ciona* single-cell sequencing data shows that *KH.L96.34* and *KH.L22.27* are sequentially expressed in notochord cells at the initial tailbud and late tailbud stages. (**B**) *Brachyury* OE driven by promoters of *KH.L96.34* and *KH.L22.27* causes no lumen formation. The yellow asterisks “*” indicate *Brahyury* OE cells. The purple represents the cell membrane boundary, and green represents the nuclei with *Brachyury* OE expression and cytoplasm with GFP expression. (**C**) Caveolin dispersed in the cytoplasm of *Brachyury* OE cells while accumulating at the apical membrane in the control group. The yellow asterisk “*” indicates *Brahyury* OE cells, yellow arrowheads indicate caveolin localization. The orange, purple, and green represent the cell membrane boundary, caveolin localization, and nuclei with *Brachyury* OE expression, respectively. The schematic images on the right side illustrate the phenotypes of the transgenic notochord cells; the numbers at the bottom left of the images indicate the percentage of notochord cells with the illustrated phenotype. All of the scale bars represent 10 μm.

**Figure 3 ijms-25-03413-f003:**
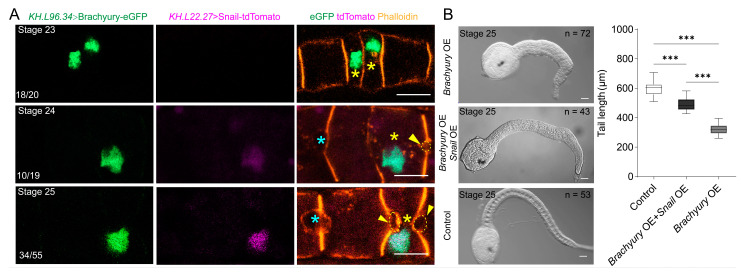
*Ci-Snail* rescues MET failure caused by *Brachyury* OE. (**A**) No visible lumen between adjacent notochord cells was present in embryo with only *Brachyury* OE at stage 23, small lumen in embryo with strong *Brachyury* OE but weak *Ci-Snail* OE at stage 24, enlarged lumen in embryo with both strong *Brachyury* OE and *Ci-Snail* OE at stage 25. The numbers at the bottom left of the images indicate the percentage of notochord cells with illustrated phenotype. The yellow asterisks “*” indicate rescued cells, yellow arrowheads and dashed lines indicate the location of the rescued lumen and the cyan asterisks “*” indicate the wild-type lumen. (**B**) Images of embryos and the tail length comparison of *Brachyury* OE, *Brachyury* OE+*Snail* OE, and control group. The numbers in the upper right of the DIC images indicate the quantity of the embryos counted. Statistical analysis was performed using *t*-tests. “***” means the difference between the two groups is extremely significant (*p* < 0.001). All of the scale bars represent 10 μm.

**Figure 4 ijms-25-03413-f004:**
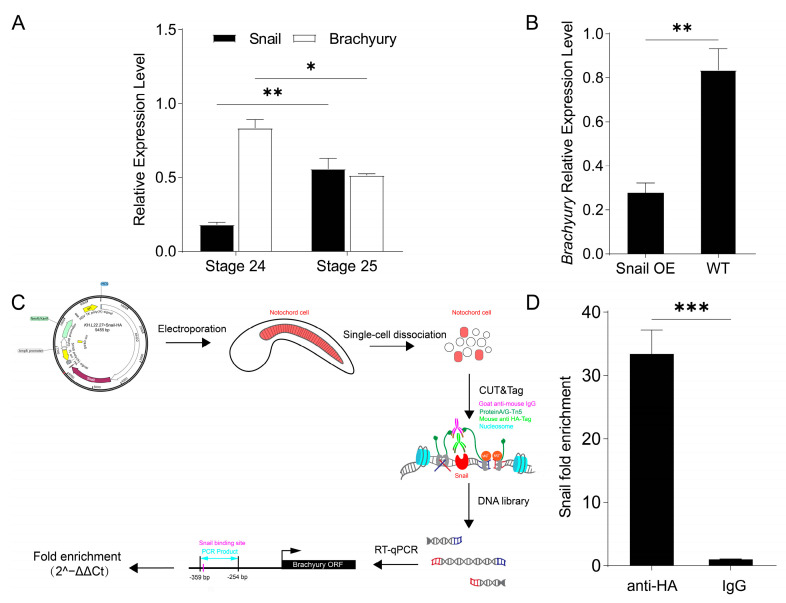
*Ci-Snail* repress*es Brachyury* expression through direct binding. (**A**) Relative expression levels of *Ci-Snail* and *Brachyury* in wild-type embryos at Stages 24 and 25. (**B**) Relative expression levels of *Brachyury* were detected in *Ci-Snail* OE and wild-type (WT) groups. (**C**) Flow diagram of CUT&Tag-qPCR method. (**D**) Fold enrichment of *Ci-Snail*-HA incubated group and control group. Statistical analysis was performed using *t*-tests. “*” indicates significant difference (*p* < 0.05), “**” indicates highly significant difference (*p* < 0.01), “***” indicates extremely significant difference (*p* < 0.001).

**Figure 5 ijms-25-03413-f005:**
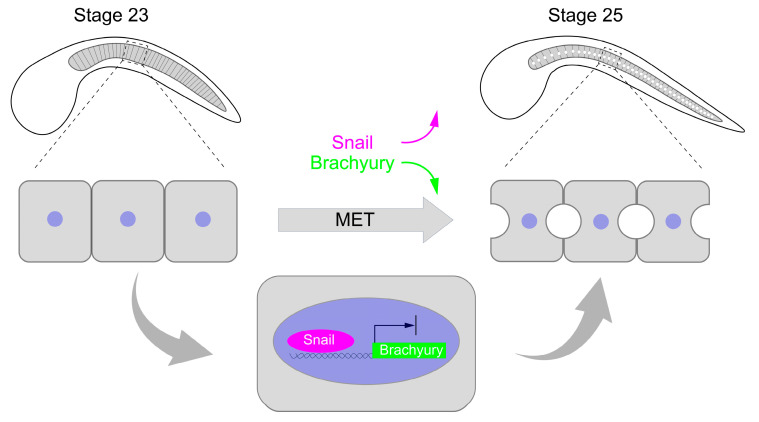
Working model of MET regulation in notochord cells. As development proceeds, *Ci-Snail* expression is up-regulated to repress *Brachyury* expression, which leads to MET. The purple represents the Snail protein, green represents the *Brachyury* gene, and light blue represents the nuclei.

## Data Availability

Data are contained within the article and Appendix A.

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
