# Peer review of "Snail Transcriptionally Represses Brachyury to Promote the Mesenchymal-Epithelial Transition in Ascidian Notochord Cells"

_ijms, 2024, doi:10.3390/ijms25063413_

Round 1
Reviewer 1 Report
Comments and Suggestions for Authors
This manuscript deals with regulatory mechanisms of Mesenchymal-Epithelial Transition in Ciona.
The submitted manuscript is scientifically sound, the methods are adequate and the results should deserve publication.
However, there are also some issues that lower the quality of this interesting work and the authors should carefully deal with them before the manuscript can be accepted for publication.
In particular, the presentation of some images should be reorganized, the Results as well as some Methods should be better described and the Introduction and the Discussion need to be broadened including relevant citations on what is already known on some regulatory mechanisms in Ciona, the genome and the chromosome complement of the study species and the possible role of heterochromatin in genomic diversification, gene transcription and expression (see below).
The Results should be reworked. In fact, it is sometimes unclear if the authors are mentioning results obtained during this study, citing previous research, describing the methods used or discussing the results. To be more clear, the authors should remove citations and several statements that are not a description of the result obtained. These citations and statements, when not strictly necessary, should be moved from the Results to the Introduction, Methods or the Discussion section, where more appropriate.
Figures are of high quality, but some of them (Fig. 1B and C, Fig. 21B and C and Fig 3A and B) should be enlarged to make them more reader friendly.
It would also be very useful to specify in the Introduction or the Discussion that heterochromatin content may have a significant role in genome and karyotype diversification as well as in transcription and expression mechanisms (Mezzasalma et al. 2019 Salamandra 55(2):140-144; Penagos-Puig and Furlan-Magaril 2020 Frontiers in cell and developmental biology, 8, 579137).
Some line-specific points are listed here below.
Abstract
Lines 13-14. Please, reword the sentence
Intro
Lines 32-35: Plese, briefly describe what is known about these processes in Ciona.
Line 55. in which stages, please be more specific
Line 57. change “developed” with “develop”
Line 58. Remove “embryo”.
Results
Lines 81-84. Please add relevant citations
Lines 128-130. These are not results. Please move the sentence in the Introduction or the Discussion.
Line 159: I am not sure that “suspected” here is appropriate.
Lines 171-175. This seems to me a discussion of the results obtained.
Lines 188-189. This is repeated from the previous paragraph.
Lines 201-207. This seems a description of the experimental design which should go in the Methods.
Discussion
Lines 227-230. Please cite your mentioned previous work.
Lines 251-267. Please at least mention that heterochromatinization may have a significant role in genome and karyotype diversification as well as in transcription and expression mechanisms (Mezzasalma et al. 2019 Salamandra 55(2):140-144; Penagos-Puig and Furlan-Magaril 2020 Frontiers in cell and developmental biology, 8, 579137)
Methods
Line 293. Please, specify the number of samples used and sampling procedures.
Line 305-310. Please, specify alignment length and procedures (e.g. ClustalW etc.)
Lines 312-328. Please, specify parameters of all the PCRs or provide relevant references
Comments on the Quality of English Language
Some sentences are a bit confusing and a minor spell check is required.
Reviewer 2 Report
Comments and Suggestions for Authors
This work reports opposite roles between Snail and Brachyury for notochord lumen formation in Ciona embryos. There are several issues that should be addressed to improve the manuscript.
1. The species of Ciona used in the study should be indicated.
2. The proportion (efficiency) of Cas9-expressing notochord cells is unclear. If electroporation was performed at 1-cell stage, why only few notochord cells express Cas9?
3. Brachyury promoter used to drive Cas9 expression is confusing. It is 1 kb in the results section but 3 kb in the methods section.
4. Why some snail-knockout notochord cells completely fail to vacuolize (figure 1B), while others are still able to vacuolize, albeit to a lesser extent (figure 1C)? This should be explained.
5. Gene names for KH.L96.34 and KH.L22.27 need to be provided.
6. Lines 145-146, the sentence “Taken together, downregulation of Brachyury expression during development facilitates notochord cells MET” seems to be in contradiction with the statement “In Brachyury-deficient Ciona embryos, notochord cells differentiation is severely impaired (line 129-130).
7. The authors should explain how overexpression of Snail partially rescues defective notochord lumen formation caused by overexpression of Brachyury, because in this condition, Snail can only inhibit transcription of the endogenous Brachyury.
8. In figures 1 and 2, unaffected cells at stage 25 form large vacuoles. However, in figure 3, unaffected cells at the same stage form relatively small vacuoles. I wonder whether embryos in different experiments were correctly staged. In addition, comparison of lumen formation in figure 3A should be done at the same stage.
9. There is no control at stage 23 in figure 3A. It is unclear whether and how notochord cells vacuolize at this stage. As I can see, unaffected cells at stage 23 do not form lumen.
10. For CUT&Tag-qPCR, the authors state that “fragmented DNA was retrieved for subsequent RT-qPCR to amplify products containing Ci-Snail binding sequences”. How do they perform reverse transcription of the DNA library in this method?
11. There is no description of immunofluorescence method.
12. The authors need to polish the writing and proof edit the manuscript, including the title main text and figure legends.
Comments on the Quality of English LanguageThere are grammatical issues in the manuscript.
Round 2
Reviewer 1 Report
Comments and Suggestions for Authors
Dear editor and authors,
I think that a good job was done by the authors in checking and correcting different sections of the manuscript. The revised version feels more readable and complete and it may be accepted for publication in my opinion.
There is still a number of small errors which can be corrected at this stage or during proof revision.
Here below are some examples.
line 21: change “another” to “the”
line 27: change “the Ci-snail overexpression” to “overexpression of Ci-snail”
line 203: I think there is a missing verb here.
line 270: change “played” to “plays”
line 304: change “evidences” to “evidence”
line 548, citation 30: change “Marcello, M.; Franco, A.; Frank, G.; M., G. F.; Gaetano, O.; Agnese, P.; Orfeo, P.” to Mezzasalma, M.; Andreone, F.; Glaw, F.; Guarino, F.M.; Odierna, G.; Petraccioli, A.; Picariello, O”.
line 550 citation 31: page number is missing.
Comments on the Quality of English LanguageMinor spell check is still required
Author Response
Point-by-point response to comments and suggestions from reviewer 1
I think that a good job was done by the authors in checking and correcting different sections of the manuscript. The revised version feels more readable and complete and it may be accepted for publication in my opinion.
There is still a number of small errors which can be corrected at this stage or during proof revision.
Here below are some examples.
line 21: change “another” to “the”
Response: Done.
line 27: change “the Ci-snail overexpression” to “overexpression of Ci-snail”
Response: Done.
line 203: I think there is a missing verb here.
Response: Yes, we replace "formation" with "formed".
line 270: change “played” to “plays”
Response: Done.
line 304: change “evidences” to “evidence”
Response: Done.
line 548, citation 30: change “Marcello, M.; Franco, A.; Frank, G.; M., G. F.; Gaetano, O.; Agnese, P.; Orfeo, P.” to Mezzasalma, M.; Andreone, F.; Glaw, F.; Guarino, F.M.; Odierna, G.; Petraccioli, A.; Picariello, O”.
Response: Done.
line 550 citation 31: page number is missing.
Response: We added page number to the cited literature.
Comments on the Quality of English Language
Minor spell check is still required
Response: Thank you for pointing this out, we have made a thorough spell check of the manuscript.
Reviewer 2 Report
Comments and Suggestions for Authors
My comments have been addressed adequately.
Author Response
Thank you very much.